# Real-Time Recognition Method for Key Signals of Rock Fracture Acoustic Emissions Based on Deep Learning

**DOI:** 10.3390/s23208513

**Published:** 2023-10-17

**Authors:** Lin Sun, Lisen Lin, Xulong Yao, Yanbo Zhang, Zhigang Tao, Peng Ling

**Affiliations:** 1Hebei Green Intelligent Mining Technology Innovation Center, Tangshan 063210, China; sunlin@ncst.edu.cn (L.S.); crazythesen@gmail.com (L.L.); fzdn44444@163.com (Y.Z.); hnlp87@163.com (P.L.); 2College of Artificial Intelligence, North China University of Science and Technology, Tangshan 063210, China; 3College of Mining Engineering, North China University of Science and Technology, Tangshan 063210, China; 4School of Mechanical and Architectural Engineering, China University of Mining and Technology-Beijing, Beijing 100083, China; taozhigang@cumtb.edu.cn; 5State Key Laboratory for Geomechanics and Deep Underground Engineering, Beijing 100083, China

**Keywords:** short-time Fourier transform, Mel spectrum, convolutional networks, acoustic emission, signal recognition

## Abstract

The characteristics of acoustic emission signals generated in the process of rock deformation and fission contain rich information on internal rock damage. The use of acoustic emissions monitoring technology can analyze and identify the precursor information of rock failure. At present, in the field of acoustic emissions monitoring and the early warning of rock fracture disasters, there is no real-time identification method for a disaster precursor characteristic signal. It is easy to lose information by analyzing the characteristic parameters of traditional acoustic emissions to find signals that serve as precursors to disasters, and analysis has mostly been based on post-analysis, which leads to poor real-time recognition of disaster precursor characteristics and low application levels in the engineering field. Based on this, this paper regards the acoustic emissions signal of rock fracture as a kind of speech signal generated by rock fracture uses this idea of speech recognition for reference alongside spectral analysis (STFT) and Mel frequency analysis to realize the feature extraction of acoustic emissions from rock fracture. In deep learning, based on the VGG16 convolutional neural network and AlexNet convolutional neural network, six intelligent real-time recognition models of rock fracture and key acoustic emission signals were constructed, and the network structure and loss function of traditional VGG16 were optimized. The experimental results show that these six deep-learning models can achieve the real-time intelligent recognition of key signals, and Mel, combined with the improved VGG16, achieved the best performance with 87.68% accuracy and 81.05% recall. Then, by comparing multiple groups of signal recognition models, Mel+VGG-FL proposed in this paper was verified as having a high recognition accuracy and certain recognition efficiency, performing the intelligent real-time recognition of key acoustic emission signals in the process of rock fracture more accurately, which can provide new ideas and methods for related research and the real-time intelligent recognition of rock fracture precursor characteristics.

## 1. Introduction

Acoustic emissions are a phenomenon in which rock deformation or fracture suddenly releases energy in the form of stress waves under external forces. It can effectively reflect the internal fracture events and fracture evolution process of rock and has become one of the most widely used monitoring and warning methods in the field of geotechnical engineering. The identification of acoustic emission precursor characteristics in rock fracture catastrophes is a major research direction in disaster monitoring and early warning in the field of geotechnical engineering. However, there is currently a lack of effective real-time identification methods for precursor characteristic signals, resulting in the poor real-time performance of precursor characteristic identification, greatly reducing the effectiveness of engineering field applications.

Acoustic emissions (AE) technology plays a vital role in identifying the precursor information for rock fractures, making it an essential tool in rock engineering monitoring and early warning systems. Previous studies have explored the correlation between AE events, the rate of strain energy, and the loading rate, establishing fluctuation trends [1]. Moreover, researchers have identified a strong association between AE energy and the evolution of rock damage [2,3,4]. The isokinetic probe using the acoustic emissions monitoring function has shown good performance in studying the characteristics of pectin films in different phase states [5]. The failure state and degree of rock formation can be analyzed using AE monitoring and energy release during the whole deformation process of sandstone under uniaxial compression [6]. Research on the frost resistance of concrete is time-consuming. The acoustic emission method can help monitor structural changes in the frost resistance process and complete the monitoring of the frost resistance of concrete from the internal structure [7]. In terms of biological monitoring, the shape and structure of acoustic emissions can be used to determine the types of lesions and deformation on the surface of key human cartilage [8]. By using acoustic emissions technology to establish a real-time monitoring system through the relationship between the acoustic emission rate and displacement rate, landslide behavior can be evaluated [9]. While these findings highlight the potential of AE technology, challenges remain. Subjective parameter selection and susceptibility to monitoring conditions, external noise, and vibrations have resulted in reduced accuracy. To address these challenges, recent advancements in signal processing techniques have focused on the time–frequency analysis of AE signals. The frequency features exhibit uniqueness, distinctiveness, and stability, allowing them to reflect various degrees of internal rock damage [10]. The acoustic emission waveform generated in rock burst events can be used to predict these events by analyzing different spectrums and peak shapes [11]. In the friction research of the sheet metal forming process, wavelet packet transform technology can be applied to AE signal processing to analyze the damage mechanism in the sheet metal forming process [12]. By analyzing the time–frequency evolution characteristics and exploring the correlation of time–frequency domain feature changes, it is possible to effectively discriminate rock fracture precursor information [13,14,15,16]. However, the application of time-frequency analysis in engineering scenarios has been hindered by computational limitations and slow recognition efficiency, limiting the applicability of real-time warning analysis. To overcome this, a novel approach based on the energy contribution rate has been proposed [17]. This approach identifies key signals that correspond to fracture events and closely characterize fracture precursor features. These key signals possess a small quantity, high energy, and a strong relationship with rock fracture precursor information. Recognizing these “key signals” addresses the challenges associated with the time–frequency analysis of large data volumes, facilitating effective precursor feature analysis and real-time warning technology implementation. Furthermore, the application of deep learning methods, particularly convolutional neural networks (CNNs) [18,19,20,21], has garnered significant interest in the field of signal recognition [22]. These approaches have shown promising outcomes in the identification of AE signals, including mining micro-seismic and blasting signals, as well as the classification of noise in composite material detection signals [23,24,25]. The deep neural network recognition method shows the ability of nonlinear mapping and effectively uses the context information of variable length [26], making them highly suitable for accurately identifying acoustic emission signals preceding rock fractures.

To address these challenges, this study proposes an innovative approach that combines acoustic emission time–frequency feature extraction with deep learning for signal analysis. The aim is to establish an intelligent model that is capable of the real-time recognition of key signals during the monitoring process, enabling the automatic identification of these crucial signals. By conducting acoustic emission monitoring experiments under uniaxial mechanical loading on granite, signal data can be collected and appropriately labeled. Various data processing techniques, including data filtering, feature extraction, and data normalization, are applied to process data and generate informative time–frequency feature images. The extracted features are then utilized to train convolutional neural networks, fine-tuning the model’s weights and other hyperparameters to enhance its accuracy. The experimental results demonstrate that the developed intelligent signal recognition model can effectively extract key signals in real-time during the signal acquisition process. It effectively addresses the issue of data redundancy in the pre-warning of a rock disaster, significantly improving the efficiency of data analysis. Additionally, it has the capability to monitor rock fissures in real-time with high accuracy. This enhances the real-time monitoring of potential disasters caused by rock destruction in underground sites, such as mines, aiming to minimize casualties and economic losses. Furthermore, it provides a scientific and convenient method for monitoring rock instability and disaster pre-warning through acoustic emissions.

## 2. Methods for Acoustic Emission Feature Extraction

### 2.1. Basic Principles of Rock Fracture Acoustic Emission Monitoring

Rocks fracture internally under external forces, and this fracture source releases energy in the form of elastic waves, which eventually propagate to the surface of the rock specimen, causing surface displacements that can be detected by acoustic emission sensors. The sensor converts the mechanical vibration on the surface of the test piece into an electrical signal, which is amplified by an amplifier and collected and processed using the acoustic emission acquisition system. Figure 1 is a basic principle diagram of acoustic emissions monitoring for rock fracture.

### 2.2. Feature Extraction Based on Short Time Fourier Acoustic Emission

In order to achieve the feature extraction of acoustic emission signals and avoid the loss of signal information, it is necessary to perform time–frequency analysis on sequence signals. Short-time Fourier transform (STFT) [27] is a time-frequency method with good characteristics in time and frequency that can be used to analyze the instantaneous variation characteristics of acoustic emission signals.

In this study, each AE signal was sampled with 2048 points. To achieve a balance in resolution, a window length of 64 with a 50% overlap was selected. The frequency spectrum length was set to 64, and a Hanning window was applied to mitigate the edge effects [28]. By concatenating the spectra of all windows, a feature map with dimensions of 65 × 65 was obtained, as illustrated in Figure 2. Additionally, to accommodate the feature input of the convolutional neural network, the feature map was resized to 80 × 80.

### 2.3. Acoustic Emission Feature Extraction Based on Mel Spectrum

To extract distinctive distribution features from acoustic emission signals, a feature extraction method based on the Mel spectrogram is proposed [29]. The frequency spectrum of acoustic emission signals exhibits non-uniform characteristics, yet most commonly used spectrogram features employ linearly scaled axes. However, a recognition method introduced in the literature [30] combines the logarithmic Mel spectrogram with CNN, resulting in a 2.87% improvement in the recognition rate compared to the linear spectrogram. Hence, by applying filtering in the Mel frequency domain to acoustic emission signals, more discriminative signal features can be extracted.

In order to enhance the characteristics and frequency distribution of the Mel spectrum, it is necessary to interpolate the acoustic emission signal. This paper uses quadratic spline interpolation [31], which produces a smoother curve compared to linear interpolation methods. This is because, between adjacent data points, quadratic spline interpolation uses quadratic polynomials to approximate data rather than linear functions. This allows the interpolation curve to have a higher derivative continuity near adjacent data points, reducing the oscillation of the curve. A specific interpolation example is shown in Figure 3. In this paper, the length of each original signal is 2048, and the length after interpolation is 40,448, making it more consistent with the signal length of common speech signals. At the same time, the sampling rate is set to 22,050, effectively transforming the short-term signal into a stationary signal with a time length of approximately 1.83 s. When the selected Hamming window step size is 512, 80 sliding windows can be obtained. The signal energy of these windows obtained at that moment is mapped to a 3D matrix representing time and Mel frequency, resulting in a visually intuitive Mel spectrogram, as shown in Figure 4. The color of the spectrogram represents the strength of the signal at this time.

## 3. Real Time Recognition Method

### 3.1. Overview of Deep Convolutional Networks

The convolutional neural network (CNN) is a special type of feed-forward neural network. Currently, in the field of target recognition, CNN’s recognition accuracy and generalization performance are superior to other standard network models with comparable parameters in deep learning [32]. CNN contains an input layer, convolutional layer, pooling layer, fully connected layer, and output layer. The network obtains feature maps of different convolutional layers through convolution operations and trains convolution kernels and biases through the backpropagation algorithm. The calculation expression of the feature map is as follows:(1)$$h_{i}=f(h_{i-1}⊗w_{i}+b_{i})$$

The symbol *h_i_* represents the feature map of the *i*-th layer, w*_i_* represents the convolutional kernel of the *i*-th layer, the ⊗ symbol denotes the convolution operation, *b_i_* is the bias vector, and *f* represents the activation function.

CNN has exhibited exceptional accuracy in image classification tasks and has achieved significant success in diverse application domains [33,34,35,36,37]. Consequently, the thorough investigation of employing efficient CNN models for the classification and recognition of 2D feature maps, such as Mel power spectrograms, STFT spectrograms, and wavelet spectrograms of acoustic emission signals, holds considerable research value.

### 3.2. Real Time Recognition Model for Acoustic Emission Signals

In this study, we utilize the VGG16 [38] deep learning network architecture, which consists of 13 convolutional layers, 5 pooling layers, and 3 fully connected layers. The convolutional kernels in each layer have a size of 3 × 3, and the pooling layers have a size of 2 × 2. VGG16 is widely acclaimed for its advantages, such as a smaller kernel size, fewer parameters, and powerful non-linear fitting capabilities. To adapt the VGG16 architecture to spectrogram data in this study, some modifications have been made. The improved structure is shown in Table 1.

The improved VGG16 structure network has a significantly reduced number of parameters, which are reduced to less than half of the original VGG16. As shown in Table 2. This lightweight network can effectively solve problems such as model computation complexity, resource consumption, and resource constraints, enabling fast response and low-power operation.

To address the imbalance in sample classification, Focal Loss [39] is introduced after improving VGG16 to replace the traditional Cross-Entropy Loss (CE loss) [40]. Focal Loss is a loss function that is used to deal with class imbalance problems. It was originally proposed to address the foreground-background class imbalance in object detection, but can also be applied to other tasks, including binary classification problems. When combined with the lightweight features of improved VGG16, the optimized model is referred to as VGG-FL. The Cross-Entropy Loss function based on binary classification is as follows:(2)$$CE(p_{t})=-\log(p_{t})$$

In Equation (2), Pt represents the probability of the model predicting a positive class. Due to the imbalance between positive and negative samples, the effect of using the Cross-Entropy Loss function directly is poor. The solution is to add a weight factor before each category in the loss function α ∈ [0, 1] to coordinate category imbalance. Based on Equation (2), the binary balanced Cross-Entropy Loss function is obtained. Pan [40] used the loss function in decoding architecture to solve the problem of disease diagnosis:(3)$$CE(p_{t})=-\alpha_{t}\log(p_{t})$$

Based on the balanced Cross-Entropy Loss function, Focal Loss adds a regulating factor γ to reduce the weight of easy-to-classify samples and focus on the training of difficult samples, which is defined as follows:(4)$$FL(p_{t})=-\alpha_{t}{\left(1-p_{t}\right)}^{\gamma }*\log(p_{t})$$

Here, (1 − *P_t_*)*^γ^* is a regulator and *γ* ≥ 0 is an adjustable focus parameter. The following figure shows the γ ∈ [0, 5] Focal Loss curve at different values. When *γ* = 0, Equation (4) degenerates into the balanced Cross-Entropy Loss Function (Equation (3)). In the experiment, α is the value selection, which can be adjusted appropriately according to the actual situation. In this paper [39], it is considered that when *γ* increases α, it needs to be reduced appropriately, and the best effect is obtained at *γ* = 2, α = 0.25, so this value can be used as a general application.

In Figure 5, the horizontal axis represents the probability of predicting the true label (Pt), and the vertical axis represents the loss (Loss). For traditional Cross-Entropy functions, the loss values are the same for samples of varying difficulty levels while Focal Loss can adjust the focus on samples of different levels of difficulty to make the model more focused on difficult-to-distinguish samples.

As shown in Figure 6, the model construction process can be divided into three main stages: the data collection and preprocessing stage, the feature extraction stage, and the deep learning convolutional training stage. In order to examine the impact of different network structures and feature extraction methods on the model’s recognition performance, six sets of ablation experiments were conducted, including different networks and feature extraction methods.

### 3.3. Evaluation Method for Unbalanced Distribution of Signal Samples

In situations where the data distribution is imbalanced, evaluating the model’s recognition performance requires a confusion matrix to be established and the calculation of precision, recall, and F1 score [41,42,43,44,45]. Unlike traditional accuracy, these metrics offer a more comprehensive and objective assessment of the classifier’s performance.
(5)$$\mathrm{precision}=\frac{\mathrm{TP}}{\mathrm{TP}+\mathrm{FP}}$$
(6)$$\mathrm{recall}=\frac{\mathrm{TP}}{\mathrm{TP}+\mathrm{FN}}$$
(7)$$F_{1}=\frac{2*\mathrm{precision}*\mathrm{recall}}{\mathrm{precision}+\mathrm{recall}}$$

In the provided formulas, TP, TN, FP, and FN represent the counts of true positives, true negatives, false positives, and false negatives, respectively.

In this study, precision refers to the proportion of correctly predicted positive instances among all instances predicted as positive, while recall refers to the proportion of correctly predicted positive instances among all actual positive instances. These evaluation metrics exhibit a trade-off relationship within the same model. When precision is emphasized and improved, it often comes at the cost of decreased recall. Recall focuses on the identification of all positive samples and is less affected by misclassifying negative samples as positive ones. Considering the nature of this study, where the avoidance of missing key signals is crucial, the importance of recall outweighs that of precision.

## 4. Experimental Verification and Analysis of Recognition Results

### 4.1. Rock Fracture Acoustic Emission Monitoring Experiment

To validate the effectiveness of the proposed method for detecting key signals in rock acoustic emissions, a total of four uniaxial compression experiments were conducted using granite specimens as the subjects.

Experimental Setup

The experimental setup, as illustrated in Figure 7, utilized the TAW-3000 servo-controlled rock mechanical testing system. This system had a maximum axial load capacity of 3000 kN and a horizontal stress load capacity of 1000 kN, with a loading accuracy error that did not exceed 1%. The acoustic emissions system employed the high-performance PCI-2 acoustic emission monitoring systems from a renowned acoustic company in the United States. It allowed simultaneous data acquisition from up to 16 signal channels. The real-time processing of acoustic emission characteristics on each channel was performed using FPGA hardware, enabling high-speed signal processing. This system featured a 40 MHz, 18-bit A/D converter for real-time analysis and provided precise signal processing capabilities. In an acoustic emissions system, setting a sampling rate of 1 MHz represented the sampling rate of 1 M points per unit of time, which could be understood as the sampling speed. This experiment is set at 1 MHz. According to the sampling theorem [46], the maximum frequency of the signal should not exceed half of the sampling rate, and the sampling value can contain all the information of the original signal. This means that the high-frequency signal cannot exceed 500 kHz, which is the highest frequency in the STFT spectrum in this article.

Specimens

The experiments utilized commonly encountered granite specimens in their natural state, as depicted in the accompanying figure. These specimens were prepared in strict accordance with international standards for rock mechanics testing. The flatness of both ends of the specimens was controlled within 0.02 mm, and their dimensions were 50 mm × 50 mm × 100 mm.

Experimental Procedure

The uniaxial compression tests in this study were conducted using the axial displacement control method. To ensure complete contact between the specimen and the loading surface and minimize any interference caused by contact noise, a preloading of 1.5 kN was applied prior to the main loading phase. The specimens were then loaded at a rate of 0.2 mm/min until failure, and the acoustic emission system was used to monitor the fracture process of the samples in real-time. To mitigate the influence of environmental noise on the acoustic emission tests, the preamplifier threshold (gain) was set to 40 dB, and the acoustic emissions sampling threshold was set to 40 dB. The acoustic emissions instrument had a sampling frequency ranging from 1 kHz to 3 MHz with a waveform sampling rate of 1 MSPS. The pre-trigger was set to 256, and the data length was set to 2 K. A high-sensitivity R6α resonant-type acoustic emission sensor was utilized, operating within a frequency range of 35 kHz to 100 kHz. Vaseline was applied between the sensor and the specimen to enhance their coupling and minimize signaling attenuation. Prior to the commencement of the experiments, a time synchronization process was performed to ensure the precise temporal alignment of data across all data acquisition systems.

### 4.2. Data Collection Results

The collected signal datasets are presented in Table 3, which provides information on the total data volume as well as the quantities of baseline key signal data and non-key signal data obtained based on the energy contribution rate [17]. The table comprises 8 sets of acoustic emission data collected from 4 granite specimens using a dual-channel acquisition setup. At the same time, in order to better reflect the signal quantity distribution, a column diagram for the comparison of the upper and lower Y-axes is drawn on the basis of Table 3 to represent the overall distribution of the number of key signals and non-key signals in the same group of tests, as shown in Figure 8.

### 4.3. Application of Intelligent Recognition Methods

#### 4.3.1. Method Application Process

The process of predicting key signals using deep learning methods can be divided into two main stages: model development and model evaluation, as illustrated in Figure 9. During the establishment of the key signal recognition model, labeled training set signal data were utilized. These data undergo preprocessing, feature extraction, and convolutional training to generate an initial predictive model. To ensure more accurate and reliable recognition performance, the initial model is subjected to testing and evaluation using unknown test datasets. The extracted features are fed into the predictive model, and the accuracy of this model is assessed by comparing the output results with the corresponding ground truth values.

The accuracy of the model can be influenced by various factors, including network hyperparameters such as the learning rate, number of training iterations, and batch size. Additionally, the choice of optimizer and the characteristics of the dataset, such as its size and distribution, can also have an impact. Therefore, it is important to carefully adjust and select suitable hyperparameters in order to optimize the network and improve the overall accuracy of the model.

In this study, all the methods were implemented using Python 3.7. For feature extraction, Librosa and Scipy libraries were utilized to extract features such as Mel spectrograms and short-time Fourier transform (STFT) spectrograms. As for the deep learning aspect, the models were trained for 15 epochs using the SGD optimizer with a learning rate set to 0.0001. The batch size was set to 8 to ensure efficient processing. It is worth noting that the network models were constructed using the PyTorch framework, which is known for its flexibility and ease of use in deep learning applications.

#### 4.3.2. Data Preprocessing

Data preprocessing plays a crucial role in eliminating low-energy weak signals with distinctive waveform features prior to feature extraction, as illustrated in Figure 10, which showcases a subset of acquired acoustic emission signal waveforms in the experiment. Through the careful comparison and analysis of these waveforms, it becomes apparent that low-energy weak signals often exhibit a characteristic of zero amplitude toward the end. This characteristic is mathematically represented by the following formula.
(8)$$A=\frac{1}{N}\sum_{i=N+1}^{2N}\left|x(i\Delta t)\right|$$

Let the signal be denoted as *x*(*t*), where each sampling point has a time interval of ∆*t*. We defined the mean absolute value of the amplitude of the last *N* sampling points of signal *x*(*t*) as *A*. Assuming that *N* is set to 10, we could determine that the amplitudes of the last 10 sampling points of *x*(*t*) were all zero if and only if *A* equaled zero.

Based on the information presented in Table 4, it can be observed that a relatively small number of signals were retained after the filtering process, while a significant proportion of signals were removed. To ensure the integrity of critical signals and the same filtering method was applied to all key feature signals. Consequently, all crucial signals were successfully preserved, thus confirming the feasibility of employing waveform-based feature selection for the identification of low-energy weak signals.

#### 4.3.3. Feature Extraction

Figure 11 and Figure 12 depict the results of the STFT and Mel spectra obtained from two different acoustic emission signals in the experiment. From the analysis of the signal waveform, it can be observed that the highest amplitude of the three noncritical signals in Figure 11a is about 0.4 mv. Compared to Figure 11a, the high amplitude of the critical signal in Figure 12a is about 0.8–2 mv. Therefore, compared to the critical signal, the overall amplitude of the noncritical signal is usually lower.

The STFT spectra of the two signal types converted using the waveform time-frequency domain are shown in Figure 11b and Figure 12b, respectively. The color axis represents the normalized amplitude in the range of 0–0.1. It can be seen from Figure 11b that the frequency corresponding to the medium and high-intensity signals is about 100 kHz; the features are relatively concentrated in a small area, and the brightness of the feature center tends to be in the range of 0.06~0.08, which represents the relative magnitude of the amplitude in this area. When observing Figure 12b (representing the STFT spectrum of noncritical signals), its characteristic distribution is relatively scattered and distributed in all time periods. The signal strength of the feature center reaches the relative maximum value of 0.1; that is, the amplitude reaches the relative maximum value, and the maximum frequency is about 30 kHz. From the STFT spectrum, it can be seen that these two signals have obvious differences in their dominant frequency, signal strength, and signal characteristic distribution.

Figure 11c and Figure 12c, respectively, show the transformed Mel spectrum of the two signal waveforms. The color scale represents the decibel value of the signal converted from the amplitude and is used to reflect signal energy levels at different time and frequency intervals. In Figure 11c, the signal energy is relatively high in the frequency range of 0–512 Hz, and the low-frequency signal distribution is relatively rich. When the frequency exceeds 1024 Hz, the signal energy shows weak spectral characteristics, and the maximum frequency is about 4096 Hz. At this time, the signal intensity is low and only appears in a short time. Figure 12c (showing the spectrum of key signals) has a relatively stable energy distribution and similar characteristics. A relatively stable medium-intensity signal is distributed at a frequency of about 1500 Hz, corresponding to about −40 db, indicating that the key characteristic signal has higher signal strength at the same frequency. At the same time, the spectrum shows the feature that low frequency corresponds to high energy, and the feature distribution of low-frequency signals is relatively rich. Compared with Figure 11c, the feature performance is more complete, with high feature similarity and identifiability. Compared with the STFT feature map, the Mel spectrum map has a good effect on the capture of low-frequency signal features, which can extract more comprehensive signal features and better reflect the characteristics of low-energy weak signals.

In general, the STFT and Mel spectrum can characterize the waveform of acoustic emission signals and can distinguish the key features and noncritical features of two kinds of acoustic emission signals through the difference of features. The STFT spectrogram pays more attention to the absolute strength of the signal and distinguishes the signal type through absolute strength. The Mel spectrogram has a good effect on the feature extraction of low-frequency signals and can better reflect the characteristics of low-energy weak signals. These features can be automatically analyzed and identified using deep learning methods to determine their signal type.

#### 4.3.4. Deep Learning Convolutional Training

In terms of data set selection and division, the selected division rules are shown in Table 5, in which the ratio of the training set and verification set is set to 5:1, and the ratio of positive and negative samples is 1:4.

Figure 13a depicts the accuracy curves of the six models on the validation set as the number of training iterations increases. It is evident that the Mel+VGG-FL model demonstrates a rapid convergence, reaching a plateau of approximately 89.3% in accuracy after 15 epochs. The final accuracy remains stable at around 87.91%, indicating that the Focal Loss technique effectively enhances the overall accuracy of the samples, even in the presence of imbalanced data. On the other hand, the accuracy curve of the Alex model exhibits a relatively lower performance with slight fluctuations. This analysis suggests that VGG16 outperforms Alex in terms of faster convergence and higher accuracy, owing to its utilization of smaller convolutional kernels, reduced parameter sharing, deeper network architecture, and improved initialization strategies.

Figure 13b illustrates the training durations of the three network models. The traditional VGG16 model, due to its larger number of network parameters, requires the longest training time. By contrast, the proposed model achieves a shorter training duration while maintaining a high level of accuracy, thereby surpassing the other comparative models.

### 4.4. Recognition Effect Evaluation

To further investigate the performance of the six models in recognizing key signals, a validation was conducted using an unseen acoustic emission test dataset. Figure 14 illustrates the distribution of recognition results on the confusion matrix for different models, while Table 6 presents the final evaluation metrics derived from the confusion matrix. The results demonstrate that all four models achieved an overall accuracy of over 79%, indicating a certain level of recognition effectiveness. However, relying solely on accuracy as a metric, it may lack persuasiveness in cases of imbalanced data distributions. Among the models, Mel+VGG-FL exhibits the highest recall rate of 81.05%, making it the preferred choice as the optimal recognition model in this study. STFT+VGG-FL demonstrates higher accuracy and precision rates, suggesting a tendency to correctly identify negative class samples (non-key signals) while occasionally missing some positive class samples (key signals). Its recognition performance is slightly below that of Mel+VGG-FL. These comprehensive metrics indicate that VGG-FL outperforms the traditional VGG16 model. Additionally, the results confirm that using Mel spectrograms as a feature extraction method is more suitable than employing STFT, while the two sets of models utilizing the AlexNet network show a slightly inferior performance compared to VGG16.

In addition, in order to further reflect the advantages of the improved vgg16 network model proposed in this paper to identify key characteristic signals, the STFT dominant frequency feature extraction combined with the clustering method [10], CNN combined with the waveform method [11], ANN combined with the traditional parameter analysis of acoustic emission [13], as well as the more commonly used network models with significant performance in recent years, were compared. RestNeSt [47], EfficientNet-v2 [48], Desenet [49], etc., are widely used. As the above has verified that the use of Mel spectrum features has a better recognition effect, the input features used in this comparative experiment are the results obtained using the Mel feature extraction method, except for the first three feature inputs combined with their own papers. In order to evaluate the final performance of different network models more objectively and accurately, the average recall rate and F1 score were used as the final recognition results. It should be noted that the network input characteristics of the comparative experiment come from the acoustic emission data monitored in this experiment. Therefore, its accuracy and running time could be affected by different input parameters. The parameter comparison of the network model is added to the experiment alongside the influence of time spent on processing each image in this experimental environment.

Based on Table 7, it can be found that, compared with traditional methods, deep learning methods such as CNN perform better in identifying the two indicators of key characteristic signals. The accuracy of identification is relatively high by introducing efficient net-v2-l and Mel input, with a recall of 79.35% and an F1 score of 77.42. However, its operation efficiency is also relatively low, and the training duration is the longest, which needs to be further optimized. While using traditional signal recognition methods, including clustering and Ann combined with acoustic emission parameter analysis, although this model has a faster running speed, the recognition accuracy is low. Considering the recognition effect of the model, its accuracy is the most worthy of attention. Summing up the above comparative experimental results, it can be seen that the running time of the proposed model is relatively general, and the operation efficiency needs to be improved; however, this method has relatively high accuracy and can accurately complete the task of the real-time identification of key characteristic signals.

Therefore, the improved VGG16 model, combined with Mel Spectrogram feature extraction methods, exhibits good applicability in key signal recognition. Further optimization can be explored, such as adjusting the window length and step size of short-time Fourier transform. These aforementioned models demonstrate a certain level of recognition effectiveness for key signals, highlighting the research value of employing deep neural network methods in the field of rock signal feature recognition.

## 5. Conclusions

In response to the challenge of low efficiency and real-time warning difficulties in recognizing key signals from acoustic emission monitoring data during rock fracturing processes, a novel approach based on time–frequency feature extraction and deep learning is proposed. This method aims to address the issue of signal redundancy and improve the accuracy of key signal identification.

This method uses two kinds of spectrums to extract signal features. Both feature extraction methods can map the original waveform features. Compared with the STFT feature map, the Mel spectrum map can extract more comprehensive signal features and pay more attention to the low-frequency feature distribution of the signal. For low-energy weak signals, it has a more comprehensive characteristic performance and effectively reduces the influence of absolute sound intensity on signal recognition. The improved VGG16 network is used for key signal recognition. Firstly, combination experiments of these two feature extraction methods and other classical neural networks are compared, and it was verified that the Mel spectrum is slightly stronger than the STFT spectrum. In order to further verify the superiority of the improved model in the identification of key features of acoustic emission and compare the identification efficiency and accuracy with other acoustic emission signal recognition methods, a series of reliable ablation experiments are carried out. The model Mel+VGG-FL in this paper has certain advantages in recognition speed, and the recognition accuracy is better than other models.

By effectively extracting key information from acoustic emission monitoring data, a solid foundation was laid for real-time early warning technology and rock fracturing. However, it should be noted that the current research only focuses on real-time key signal identification in the uniaxial fracturing tests of granite. This method needs to be further applied to different experimental scenarios to enhance its versatility and broaden its applicability. Rock disasters on the construction site are characterized by suddenness and unpredictability. It can easily cause casualties and engineering losses. This article proposes a more intelligent method for identifying the key characteristics of rock rupture precursors based on the shortcomings of real-time monitoring. After being improved, it is expected that this can be applied in engineering practice and provide a relatively convenient and intelligent strategy for the early warning of rock disasters.

## Figures and Tables

**Figure 1 sensors-23-08513-f001:**
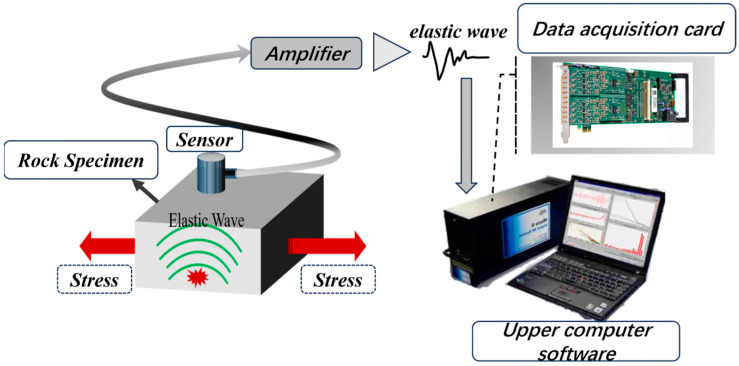
Basic principles of rock fracture acoustic emissions monitoring.

**Figure 2 sensors-23-08513-f002:**
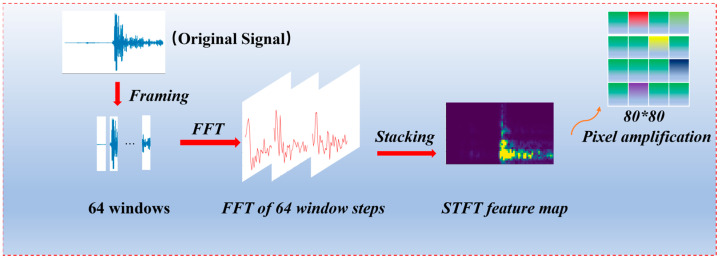
Generation of STFT feature maps for acoustic emission signals.

**Figure 3 sensors-23-08513-f003:**
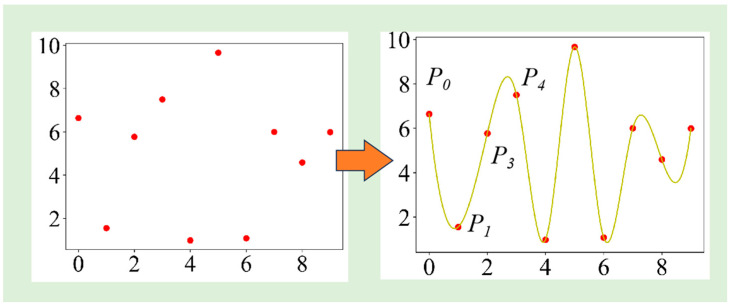
Quadratic spline interpolation.

**Figure 4 sensors-23-08513-f004:**
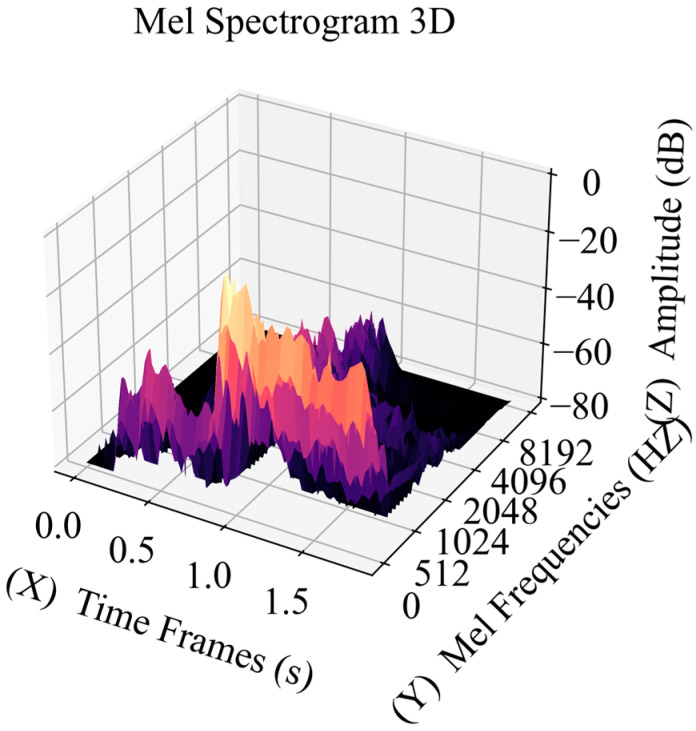
Mel spectrum feature map, where the highest Mel frequency is 8192 Hz, with a duration of nearly 2 s and a decibel setting of −80~0.

**Figure 5 sensors-23-08513-f005:**
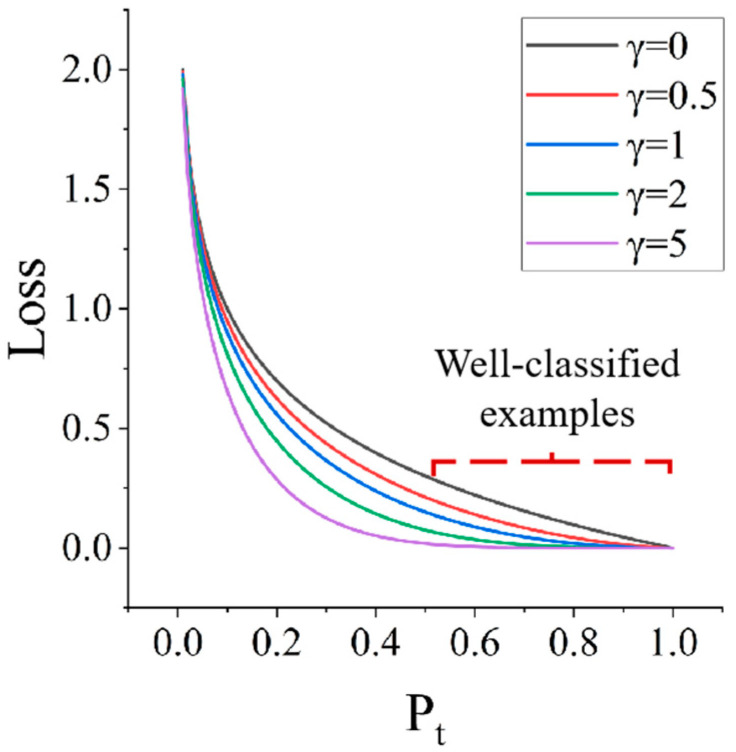
Different γ Focal Loss curves of value γ control the shape of the curve γ. The larger the value, the smaller the loss of good classification samples, and we can focus our model’s attention on those difficult to classify samples.

**Figure 6 sensors-23-08513-f006:**
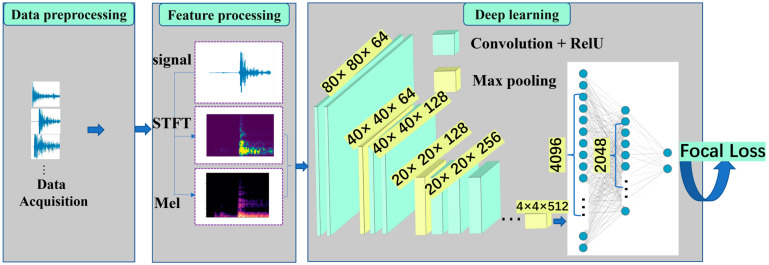
Model building process.

**Figure 7 sensors-23-08513-f007:**
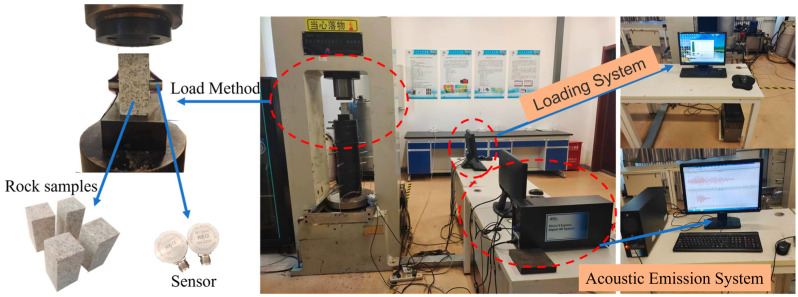
Test equipment and samples.

**Figure 8 sensors-23-08513-f008:**
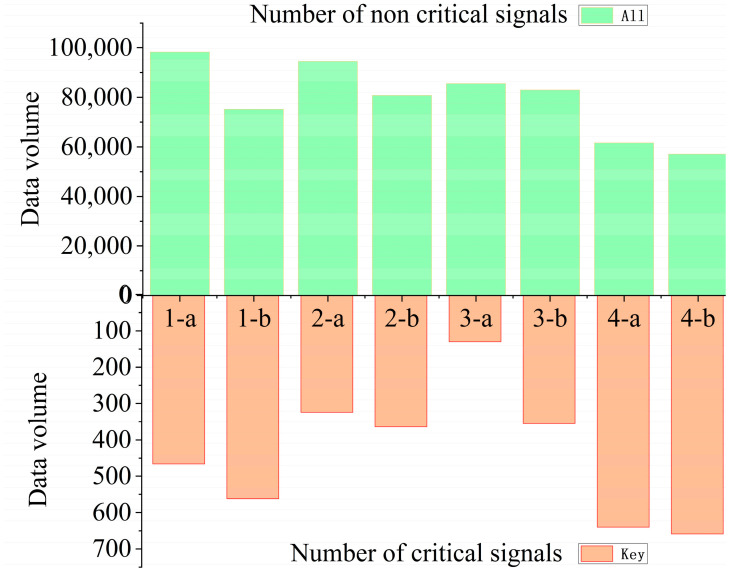
Comparison of the number of two types of characteristic signals in each test.

**Figure 9 sensors-23-08513-f009:**
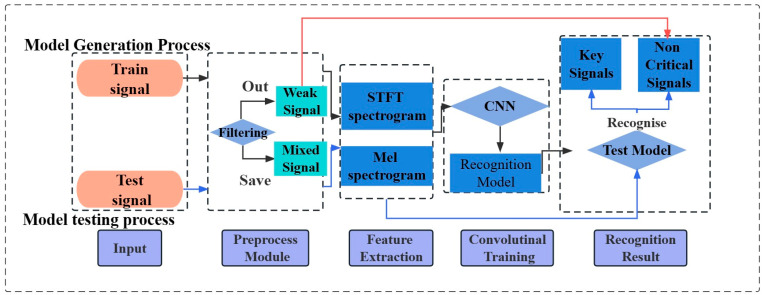
Integrated framework for feature extraction and deep learning. The construction of this framework requires a large number of signal datasets with known labels.

**Figure 10 sensors-23-08513-f010:**
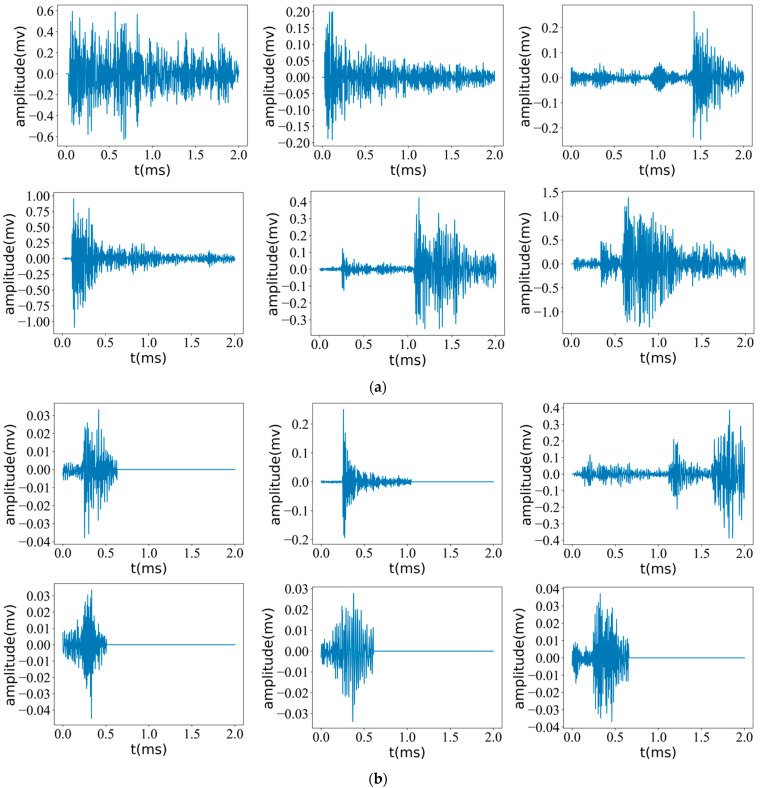
Rock key characteristic signal and low energy weak signal waveform (**a**): Low energy weak signal (**b**): Key signals. Each signal had a duration of 2 ms, and 6 groups were randomly selected from each of the two signal classes for observation.

**Figure 11 sensors-23-08513-f011:**
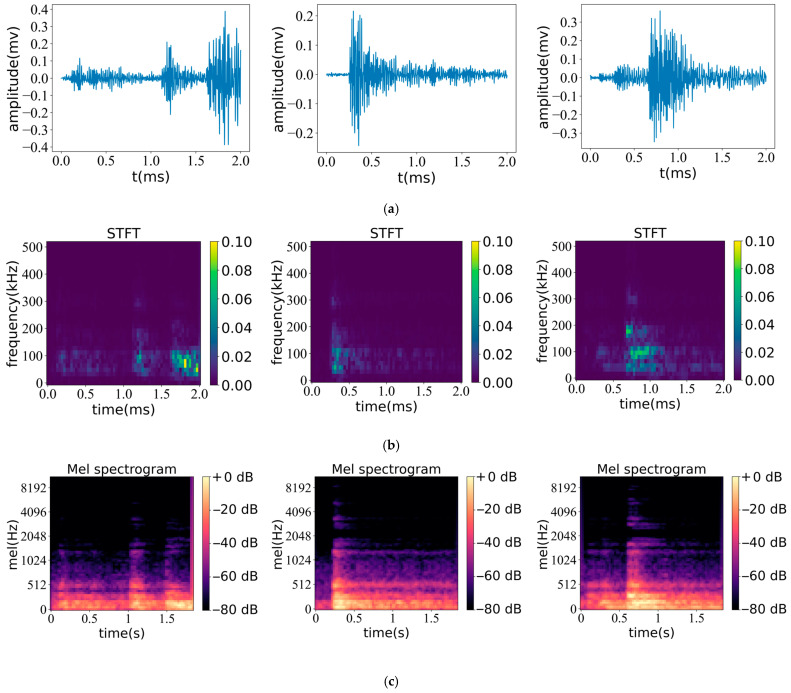
Non critical signal waveform and corresponding spectrum diagram: (**a**) Signal waveform; (**b**) STFT spectrogram; (**c**) Mel spectrogram.

**Figure 12 sensors-23-08513-f012:**
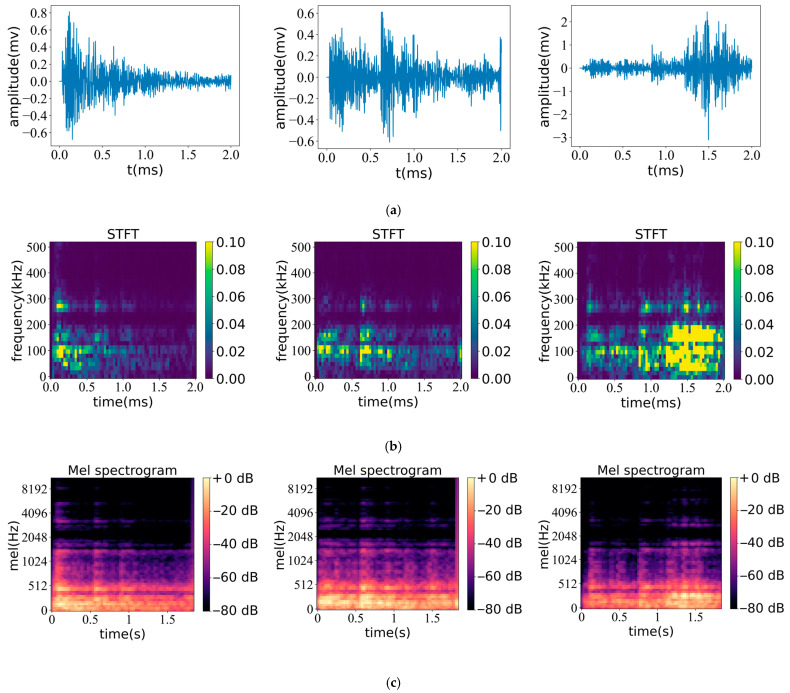
Key signal waveform and corresponding spectrum diagram: (**a**) Signal waveform; (**b**) STFT spectrogram; (**c**) Mel spectrogram.

**Figure 13 sensors-23-08513-f013:**
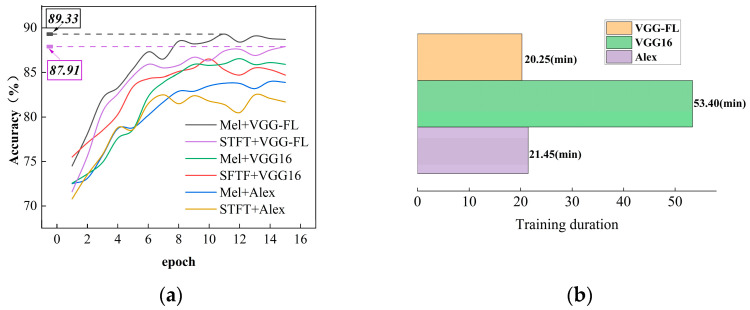
The results and cost of the validation set during the training process: (**a**) Accuracy curves, (**b**) Training duration. The accuracy curve includes the accuracy results of six models, which are color-labeled in the figure.

**Figure 14 sensors-23-08513-f014:**
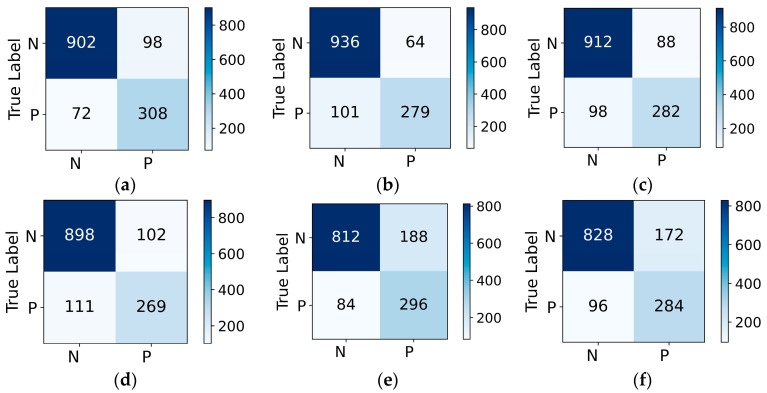
Confusion matrix of six recognition models: (**a**) VGG-FL with Mel; (**b**) VGG-FL with STFT; (**c**) VGG16 with Mel; (**d**) VGG16 with STFT; (**e**) Alex with Mel; (**f**) Alex with STFT. The horizontal axis represents the predicted data volume, and the vertical axis represents the true value.

**Table 1 sensors-23-08513-t001:** Improved VGG16 model architecture and number of features.

Network Layer	Kernel Size	Step Size	Number of Channels	Convolutional Features
Conv1, Conv2	3 × 3	1	64	80 × 80 × 64
Pool1	2 × 2	2	64	40 × 40 × 64
Conv3, Conv4	3 × 3	1	128	40 × 40 × 128
Pool2	2 × 2	2	128	20 × 20 × 128
Conv5, Conv6, Conv7	3 × 3	1	256	20 × 20 × 256
Pool3	2 × 2	2	256	10 × 10 × 256
Conv8, Conv9, Conv10	3 × 3	1	512	10 × 10 × 512
Pool4	2 × 2	2	512	5 × 5 × 512
Conv11, Conv12, Conv13	3 × 3	1	512	5 × 5 × 512
Pool5	2 × 2	1	512	4 × 4 × 512
FC1				4096
FC2				2048
FC3				2

Note: Conv is a convolutional layer; Pool is the pooling layer; FC is a fully connected layer.

**Table 2 sensors-23-08513-t002:** Parameters relating to different network models.

Network Layer	Network Parameter (M)	Loss Function
VGG-FL	56.68	Focal Loss
VGG16	138.36	Cross-Entropy Loss
Alex	61.10	Cross-Entropy Loss

**Table 3 sensors-23-08513-t003:** Acoustic emission signal dataset, according to the effective theory [17]; the signals are labeled and differentiated, and the key signals account for less.

Rock Specimens	All Signals	Key Signals	Noncritical Signals
HG-1-a	98,726	466	98,260
HG-1-b	75,809	562	75,247
HG-2-a	94,907	324	94,583
HG-2-b	81,118	364	80,754
HG-3-a	85,731	130	85,601
HG-3-b	83,378	355	83,023
HG-4-a	62,230	640	61,593
HG-4-b	57,760	659	57,101
Total	639,659	3500	636,162

**Table 4 sensors-23-08513-t004:** All low energy weak signal screening results.

Signal Type	Initial Data Volume	Filtered Data Volume	Removal Rate
Noncritical signals	636,162	98,686	84.49%
Key signals	3500	3500	0%

**Table 5 sensors-23-08513-t005:** Partition of datasets.

Name	Key Signals	Non Critical Signals
Initial training set	2600	10,400
Initial validation set	520	2600
Test set	380	1000

**Table 6 sensors-23-08513-t006:** Evaluation index based on the confusion matrix.

Network Model	Characteristic Spectrum	Accuracy (%)	Precision (%)	Recall (%)	F1 (%)
VGG-FL	Mel	87.68	75.86	81.05	78.37
	STFT	88.04	81.34	73.42	77.17
VGG16	Mel	86.52	76.21	74.21	75.19
	STFT	84.57	72.51	70.79	71.63
Alex	Mel	79.31	61.16	77.96	68.44
	STFT	80.58	62.29	74.74	67.91

**Table 7 sensors-23-08513-t007:** Comparative experimental results of various models.

Model	Input Feature	Params (M)	Infer Time (ms)	Train Time (min)	Recall (%)	F1 (%)
Cluster [10]	STFT params	-	-	6.4	67.23	62.43
CNN [11]	Waveform	24	50	13.5	73.45	71.53
ANN [13]	AE params	-	-	4.7	62.42	64.68
VGG-FL(Ours)	Mel	56	117	20.2	81.05	78.37
ResNeSt-101 [47]	Mel	48	131	28.4	75.32	76.58
EfficientNet-V2-L [48]	Mel	121	192	45.4	79.35	77.42
DeseNet-169 [49]	Mel	14	78	17.5	73.52	72.68

## Data Availability

Not applicable.

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
