# Peer review of "Real-Time Recognition Method for Key Signals of Rock Fracture Acoustic Emissions Based on Deep Learning"

_sensors, 2023, doi:10.3390/s23208513_

Round 1

Reviewer 1 Report

This manuscript provides a recognition method for key signals of rock fracture acoustic emission based on deep learning. However, the manuscript devoted a significant amount of space to introducing many well-known theories and technologies in this field, with limited introduction to the recognition method. Following are my comments:

    (1) What are the main issues to be addressed in the manuscript? real-time processing of rock fracture acoustic emission signal, or imbalanced sample classification, or anything else?

    (2) The experiment is very insufficient. There is no experimental on the running time of the proposed model, and there is no comparative experiment with other relevant methods (such as reference [10]).

    (3) The novelty is limited. Time-frequency spectral, Mel features, VGG and AlexNet are widely used in this filed.

   (4) Many of the content in the manuscript is introductory and lacks details. 

   The quality of English need improves.

Reviewer 2 Report

In equation 1 (line 133), the designation of two symbols is the same. I would distinguish between omega and e.g. w for window. On lines 115 to 117, it is not clear why the authors used w and not omega.

Line 143: I don't understand where the authors got equation 2. So why are there constants of 2595 dB (?) and 700 Hz. Similarly, equation 3.

The linear interpolation (line 154) looks quite irregular. Personally, I'd rather use some kind of polynomial or spline. The linear assignment of an extra twenty points between the measured points is somewhat illogical.

Why are the constants gamma equal to 0 and alpha equal to 2 chosen in equations 5 and 6? (lines 207-211).

A graph could also be plotted with Table 3. (291)

Which parts of Python have been created by the authors and where the code can be downloaded. (312)

I can not find the Table 5. (383)

Why is the same designation omega for different constants, variables or symbols? (113, 175)

I didn't understand the difference between key characteristic and normal.

At what frequency was the  sampling rate? If the data recording was 1 MSPS then the maximum sampling frequency is 1 MHz. (276)

Reviewer 3 Report

This article presents an innovative method based on AE signals post-processing/analysis for real-time recognition of key signals of rock fracture. The methodology is very interesting and the results are promising. This article deserves to be published since it can provide an important contribution to the State of the Art of this topic. The article is well written and its quality is high. Therefore, I recommend this article to be published after a minor revision:

* The title of Section 4.3 is wrong.

* There is a mistake in the title of Section 4.3.3, the first letter must be a capital letter.

* The Introduction is brief; more background/references about different applications of the AE should be included, for example:

(a) Monitoring of civil structures (bridges, tunnels, buildings, etc.).

e.g.: Weld acoustic emission inspection of structural elements embedded in concrete, Science Progress, 105(1), 1–20, 2022. doi:10.1177/00368504221075482

(b) Diagnosis of pressure vessels and storage containers.

(c) Detection of failures caused by cracks in aerospace structures.

(d) Investigation of material properties, failure mechanisms and damage behavior.

(e) Quality control and inspection of different processes.

(f) Detection and location of leaks in real time in tanks or buried pipes.

* The parentheses of Equation (2) are too small.

* The quality of Figure 4 must be increased and the format of the legends of the two axes must be uniform.

* In Figure 5, the orientation of the vertical legends must be uniform.

* The units of the different parameters must be written into parentheses instead of using a diagonal slash. Please re-check the entire article including figures.

* In the last part of Section 4 an additional sub-section called “Results discussion” should be included. In this sub-section, a comparative table should be presented, where the method proposed in this article is compared against other works (with references included) using similar and different techniques for the same purpose and highlighting the advantages of the proposed methodology.

Minor editing of English language required

Reviewer 4 Report

1.         Figure 1 Suggested design of composition to increase the beauty of the picture.

2.         Figure 2 The picture and title are suggested on a page.

3.         In Figure 4, 5, 7, 8, 9, 10, 11, and 12, the numbers and names of the axes are too small. You are advised to adjust them again.

4.         Some pictures have low pixel, it is recommended to export the HD original image into the article.

5.         In the abstract of this paper, there are few descriptions of the current research process of precursor signal recognition of rock fracture disaster monitoring, so it is inappropriate to directly introduce the content of this study, and it is suggested to supplement.

6.         In the introduction part, the description of the significance of this study in real life is not comprehensive enough to add relevant descriptions in this aspect.

7.         Table 3 contains basic spelling problems. You are advised to modify them.

8.         The conclusion part mainly focuses on the role of the research and the current shortcomings, and does not provide a sublimation summary. It is suggested that the author review the full paper and rewrite the conclusion.

9.         The overall structure of the article is not perfect enough. The author suggests readjugating the structure of the whole text.

10.     There are still many basic grammatical errors in the text. The author should read the whole text carefully and correct the existing basic grammatical errors in the text.

 Moderate editing of English language required

Round 2

Reviewer 1 Report

1) The manuscript still spends a lot of space introducing well-known theories in signal processing fields such as STFT and Mel frequency. It is unnecessary.

2) The authors argue that " In order to enhance the characteristics and frequency distribution of the Mel spectrum, it is necessary to interpolate the acoustic emission signal". However, interpolating digital signal will only increase the display resolution of frequency spectral. This procedure cannot bring new meaningful information to the signal. What's the purpose of the interpolation?

3) Despite the author's statement of their innovation, I feel that the innovation of the manuscript is still very limited.

Moderate editing of English language required.
